# Stutter-TTS: Controlled Synthesis and Improved Recognition of Stuttered Speech

**Xin Zhang**[*]  **Iván Vallés-Pérez**[*]  **Andreas Stolcke**  **Chengzhu Yu**
**Jasha Droppo**  **Olabanji Shonibare**  **Roberto Barra-Chicote**  **Venkatesh Ravichandran**
Amazon Alexa AI
{xizhanga, stolcke, czyu, drojasha, olabanjs, veravic}@amazon.com
{ivallesp, rchicote}@amazon.co.uk

## Abstract

Stuttering is a speech disorder where the natural flow of speech is interrupted by blocks, repetitions or prolongations of syllables, words and phrases. The majority of existing automatic speech recognition (ASR) interfaces perform poorly on utterances with stutter, mainly due to lack of matched training data. Synthesis of speech with stutter thus presents an opportunity to improve ASR for this type of speech. We describe Stutter-TTS, an end-to-end neural text-to-speech model capable of synthesizing diverse types of stuttering utterances. We develop a simple, yet effective prosody-control strategy whereby additional tokens are introduced into source text during training to represent specific stuttering characteristics. By choosing the position of the stutter tokens, Stutter-TTS allows word-level control of where stuttering occurs in the synthesized utterance. We are able to synthesize stutter events with high accuracy (F1-scores between 0.63 and 0.84, depending on stutter type). By fine-tuning an ASR model on synthetic stuttered speech we are able to reduce word error by 5.7% relative on stuttered utterances, with only minor ($< 0.2\%$ relative) degradation for fluent utterances.

## 1 Introduction

According to the National Institute on Deafness and Other Communication Disorders, nearly three million Americans suffer from lifelong stuttering. Advances in deep learning have facilitated the development of ASR systems and encourage the integration of voice assistant in various commercial products (Kepuska and Bohouta [1]). However, people who stutter by and large have not benefited from this added convenience, as existing ASR systems have difficulties understanding atypical speech, resulting in poor performance when it comes to stuttering (Barrett et al. [2]).

Recent research efforts have been dedicated to improved automatic detection and recognition of atypical speech using neural networks(Bayerl et al. [3], Jouaiti and Dautenhahn [4]). Despite advances in modeling technology, these systems are limited by a lack of relevant atypical speech data, and their performance greatly depends on sufficient stuttered speech for model training ([2]). For comparison, it is common today to build automatic speech recognition systems using at least 1,000 hours of labeled data (Panayotov et al. [5]), but the recently introduced SEP-28 dataset contains utterances with stutter comprising less than 24 hours (Lea et al. [6]).

Synthetic TTS data has already shown to be useful in improving automatic speech recognition accuracy for out-of-vocabulary words (Zheng et al. [7]), and this paper focuses on using similar technology to address the atypical data sparsity problem. As a necessary first step towards this goal,

---

[*]These authors contributed equally.

we focus here on the design of a TTS model capable of generating realistic and natural speech with diverse forms of stutter, with an initial validation of the benefit to ASR performance.

TTS technology has been widely utilized to produce artificial voices that closely emulate natural human speech (Bilinski et al. [8]). In particular, end-to-end TTS synthesis has attracted wide attention as a result of the simplification in training and improved naturalness of synthetic utterances. Recent work has demonstrated the creation of multiple voices for context-aware conversational speech synthesis (Stanton et al. [9], Cong et al. [10]). Soleymanpour et al. [11] reported on synthesis of dysarthric speech based on a multi-speaker TTS framework. To the best of our knowledge, no literature has investigated how to leverage TTS for synthesizing different types of stuttering voices. For people who stutter, the natural flow of speech is interrupted by various irregular acoustic patterns, such as sound repetition, syllable prolongation, and long pausing. Consequently, it remains a challenge to extend current TTS approaches to the production of realistic stuttering with high naturalness and fine-grained control over the location and types of stutter events.

To address these limitations, we introduce Stutter-TTS, a novel TTS approach that achieves both naturalness and natural prosody in stuttered speech synthesis. We propose a novel control strategy for supervised learning by incorporating special tokens into the source text to represent different prosodic-phonetic characteristic of stutter, including phoneme repetition, dysrhythmic phonation, and blocks. By manipulating the input source text, Stutter-TTS can generate either fluent speech (without stutter) or specific types of stutter in specific locations. We systematically produce 100 hours of diverse types of utterances containing stutter and quantify the generation performance by randomly sampling 400 utterances for evaluation.

## 2 Methods

A multi-speaker transformer-based TTS network has been used previously to model stuttering speech. The architecture is similar to Chen et al. [12] and Li et al. [13], consisting of a transformer backbone, a phonetic encoder, and an acoustic autoregressive decoder. A scaled dot product (Kamath et al. [14]) attention mechanism aligns the acoustic and phonetic features.

### 2.1 Stutter-TTS Architecture

The architecture is presented in 1. The transformer backbone comprises the phonetic encoder and the decoder, which transforms linguistic input sequences into acoustic output sequences.

To condition the decoder on speaker identity information, a global audio reference encoder is included. This module consists of a Gated Recurrent Unit (GRU) (Chung et al. [15]) network that receives a set of randomly drawn frames from a reference Mel-spectrogram, and aggregates them into a time-independent representation. This "reference embedding" is intended to encode non-linguistic information about the desired output, such as speaker identity and prosody. At training time, the input to the reference encoder is the target Mel-spectrogram. At inference time, a reference Mel-spectrogram of the desired speaker is used as a prototype of the voice to use to synthesize.

A stuttering disorder is often characterized by unintentional repetitions, prolongations, or interruption of sounds. It is very difficult to predict which phonemes in an utterance will be affected by stutter (Dash et al. [16]). From the speech generation perspective, this leads to a situation where the text-to-speech mapping is more ambiguous than for regular speakers. Following this line of reasoning, a probabilistic embedding is added as input to the phonetic encoder. Instead of modeling a constant embedding for each phoneme, the parameters of a diagonal Gaussian are used. This feature allows the model to learn the pronunciation uncertainty at phoneme level. Additionally, a learnable parameter $\alpha$ is used to weight the sum of the positional encoding, together with a layer normalization (Ba et al. [17]), which as described in Chen et al. [12], assures that both phonetic and positional information are preserved.

Autoregressive decoders, especially when dealing with data that features local correlations as found in speech, often stall into a failure mode known as exposure bias (Arora et al. [18]): the decoder, instead of predicting the next step, copies its last input step. To prevent exposure bias, a prenet with a strong regularization is included before the decoder. This module is vital for the correct generalization of the model. It consists of a strong dropout (60%) that is kept active at inference time (Gal and Ghahramani [19]) followed by a strong bottleneck projection Chen et al. [12]. This regularization reduces the

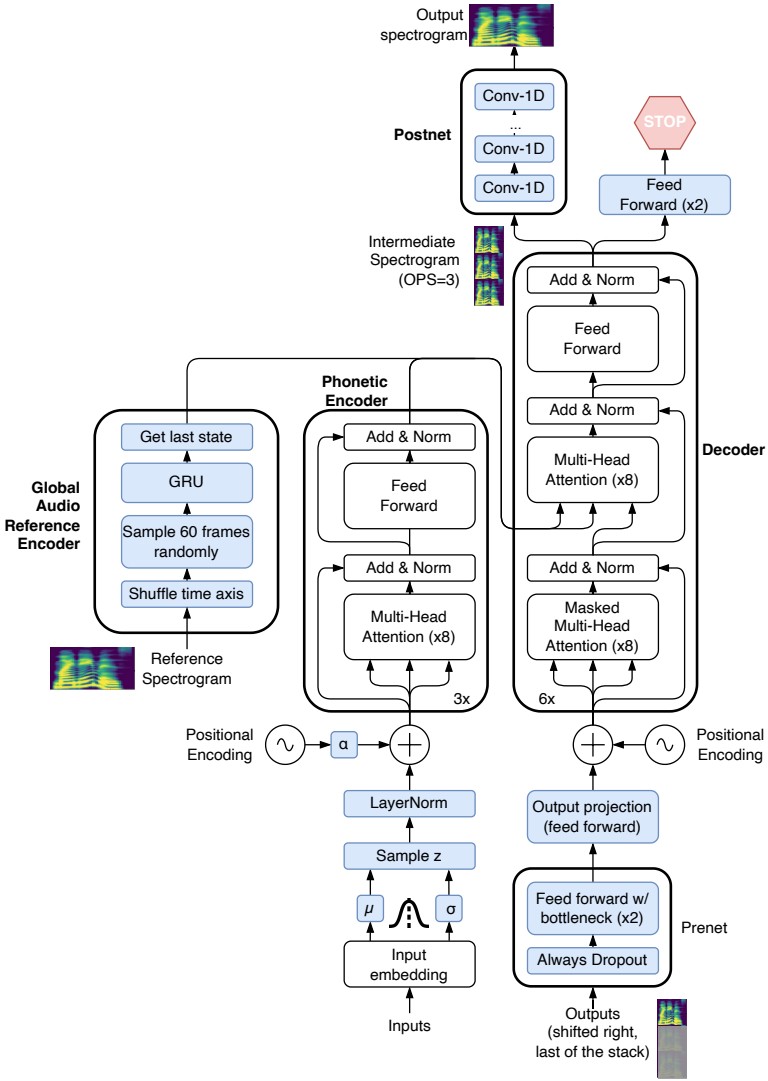

Figure 1: Diagram of Stutter-TTS architecture.

amount of information that is given to the decoder at each step, preventing it to stall into the exposure bias failure mode. Finally, after the decoder module, a postnet and a stop signal are included, similar to the Tacotron 2 architecture (Wang et al. [20]).

To train the model, the L1-loss between the target and the predicted Mel spectrogram is minimized using stochastic gradient descent, similar to the original loss function of Tacotron 2 (Wang et al. [20]). For efficiency during inference, it is necessary that the output of the decoder be a reasonable predictor of the correct spectrogram, even before passing through the postnet. Accordingly, an additional L1-loss on this intermediate output is included. Equation 1 shows the full loss function, where $\hat{\mathbf{m}}_{\text{final}}$ is the mel-spectrogram after the postnet, $\hat{\mathbf{m}}_{\text{intermediate}}$ is the mel-spectrogram before the postnet and $\mathbf{m}$ is the target mel-spectrogram. The TTS model is trained using the *teacher-forcing* method (Williams and Zipser [21], Goodfellow et al. [22]). At inference time, the free-running mode is used, generating the samples one step at a time in an autoregressive fashion. The autoregressive loop contains the decoder and the prenet modules, but not the postnet module Wang et al. [20].

$$J(\mathbf{m}, \hat{\mathbf{m}}_{\text{intermediate}}, \hat{\mathbf{m}}_{\text{final}}) = \cdot ||\hat{\mathbf{m}}_{\text{intermediate}} - \mathbf{m}||_1 + ||\hat{\mathbf{m}}_{\text{final}} - \mathbf{m}||_1 \qquad (1)$$

## 2.2 Stutter Tokens

To replicate recurring prosodic-phonetic phenomena associated with stutter, we use a list of special tokens to denote different stuttering patterns and their location. Specifically, we insert stutter tokens immediately in front of the word where stuttering occurs in the corresponding audio.[2] In this work, we mainly focus on three common stutter types as described in Table 1. During grapheme-to-phoneme (G2P) conversion, each stutter token is mapped to a corresponding special phoneme; these stutter phonemes are added to the regular phoneme set. The TTS model will hence learn embedding vectors associated with each of the stutter types.

Table 1: The mapping rule from different types of stutter to corresponding tokens inserted in the source sentence. Type frequencies in the annotated training dataset are given relative to the total number of utterances.

| Stutter type | Stutter token | Rel. frequency (%) |
|---|---|---|
| Phoneme repetition | s_repetition | 40.11 |
| Dysrhythmic phonation | s_phonation | 21.40 |
| Block | s_block | 15.59 |

As illustrated in Figure 2, stutter labels are introduced into the input sentence to denote certain prosodic-phonetic structure. It is worthwhile pointing out that the proposed processing approach achieves word-level control in terms of where stuttering happens in the synthetic utterances. This design allows fine-grained control of stutter occurrences at synthesis time. In the inference stage, we can simply place the token for the desired stuttering pattern prior to the word where we want the model to render a stutter event. The resulting synthetic audio will produce a stutter at the designated position in the source sentence.

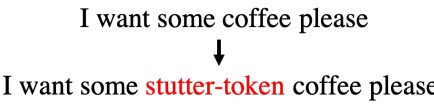

Figure 2: Encoding of stutter events in the TTS input transcript. A stutter token is inserted in the utterance precisely prior to the stuttered word. The stutter token type is chosen according to the desired type of stutter pattern.

# 3 Experimental Results

## 3.1 Dataset and Model Training

The Stutter-TTS model is trained using a combination of two internal datasets, one containing fluent speech (without stutter) captured on close-talking microphones, and one with human-produced, annotated stuttered speech. The fluent speech dataset contains 10 professional speakers with 13,000 studio-recorded utterances per speaker (600 hours in total). The human stuttering dataset is collected from speakers who exhibit stutter, in a noise-free environment via mobile phones; it contains 146 native English speakers, with 125 utterances per speaker (40 hours in total). Utterances in both datasets are 6 to 12 seconds long.

We process all audio at 16 kHz and generate 80-dimensional Mel spectrograms. The length of a frame is 50 ms with an overlap of 12.5 ms. We employ the universal neural vocoder proposed by Lorenzo-Trueba et al. [24] to synthesize audio samples from spectrograms generated by Stutter-TTS.

---

[2]Representing prosodic and structural properties with between-word tags for the purpose of word sequence modeling is quite natural and has been used in other speech modeling tasks, such as for prosody-based ASR [23].

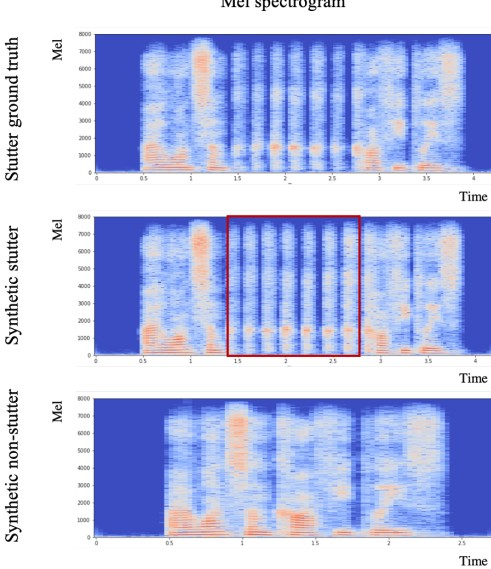

Figure 3: Comparison of Mel spectrograms of ground truth versus synthetic stuttered speech.

## 3.2 Evaluation of Synthetic Stuttered Speech

To evaluate the synthesis of utterances with stutter, we compare the Mel spectrogram generated from Stutter-TTS with the associated recording, collected from speakers with stutter. We modify the original transcription by inserting the stutter tokens where the speaker stuttered, with the aim to reproduce the stutter pattern of the original recording. One example is shown in Figure 3, where the red rectangle highlights where our model is able to mimic repetition patterns. Observe that when eliminating the stutter token from the source text, the resulting synthetic utterance contains no stuttering, thus preserving the ability to produce fluent utterances with high naturalness.

To generate a synthetic stuttered speech set, we synthesized 100 hours of speech containing all three stutter types. For each of 10 speakers, we sample 20 reference recordings and pair them with 10,000 sentences. Stutter tokens are randomly inserted into input transcripts with equal probability for all word positions. To measure generation performance with stutter, we randomly sampled 500 utterances containing phoneme repetition, dysrhythmic phonation, block and non-stutter. We evaluate the existence of specific stutter types in a subjective manner, by listening to the synthesized waveforms and identifying whether the desired stuttering patterns occur; we also excluded about 20% of samples for which the TTS had failed to produce any intelligible speech.

Table 2: F1 scores for correct synthesis of different types of stutter, while varying the ratio of fluent to stuttered utterances in training.

| Training ratio | Phoneme Repetition | Dysrhythmic Phonation | Block | Non-Stutter |
|---|---|---|---|---|
| 95:5 | 0.692 | 0.503 | 0.720 | 0.647 |
| 90:10 | **0.786** | **0.633** | 0.837 | **0.733** |
| 85:15 | 0.773 | 0.615 | **0.853** | 0.575 |

Evaluation results are summarized in Table 2, in terms of F1 scores for successful generation of the specified stutter types.[3] It is crucial for good performance to train on a combination of fluent and stuttered speech. We vary the ratio between the two types of utterances when sampling data for assembling minibatches for training; a fluent-to-stuttered ratio of 90:10 gives a good compromise

---

[3]The F1 score is the harmonic mean of recall and precision; recall is the rate at which an intended stutter is correctly realized in the audio; precision is the rate at which an observed stutter was intended based on the input specification.

between over- and under-generating stutter events, and similar F1 scores for stutter events and fluent word synthesis (ranging from 0.633 to 0.837).

### 3.3 Evaluation of ASR Performance

We also evaluate the effect of synthetic speech containing stutter on ASR training. Specifically, we fine-tuned an RNN-T-based ASR model (Zheng et al. [7]) for an additional five epochs using the 100 hours of synthetic utterances produced by Stutter-TTS, along with fluent speech sampled from a corpus of 66k hours of recorded utterances (the same data used for training the baseline model). Instead of checking for TTS failures by listening we used an automatic script to remove empty waveforms.[4] Each epoch consists of 1000 steps with minibatch sizes of 32, 16, and 8 for utterances of length up to 3, 6, and 9 seconds, respectively. During minibatch assembly for fine-tuning, we explore different sampling ratios of utterances with versus without stutter. The sampling ratio is a critical hyperparameter chosen so as to improve the ASR accuracy on stuttered speech without a substantial degradation on fluent speech. Table 3 summarizes the relative word error rate (WER) reduction at different sampling ratios, compared to the baseline RNN-T model, showing the tradeoff between WER reduction on stuttered versus fluent test speech. A good tradeoff is obtained by sampling 3% of the training data from synthetic speech with stutter, achieving 5.74% relative WER reduction for human utterances with stutter, while limiting the WER degradation on fluent speech to 0.18% relative. All WER reductions for speech with stutter are statistically significant (with $p < 10^{-6}$).

Table 3: Relative change in ASR word error rate when fine-tuning an RNN-T ASR model using different ratios of fluent to synthetic stuttered speech. The last column gives the results of a Wilcoxon rank sum test for the error reduction seen for stuttered speech.

| Training ratio | Test set with stutter | Test set without stutter | $p$-value |
|---|---|---|---|
| 99:1 | -2.78 | 2.13 | $< 10^{-6}$ |
| 97:3 | **-5.74** | **0.18** | $< 10^{-6}$ |
| 95:5 | -3.89 | 0.35 | $< 10^{-6}$ |
| 90:10 | -4.17 | 1.24 | $< 10^{-6}$ |

It is worth pointing out that less-than-perfect accuracy in synthesizing stutter events is not necessarily a problem for using Stutter-TTS in ASR training. A certain number of omitted or spurious stutter events should not affect ASR training as long as the overall distribution of event types is not too skewed, since the reference transcription used in ASR training always consists of the corresponding fluent transcript.

## 4 Conclusion

We present a novel text-to-speech system, Stutter-TTS, that can fabricate stuttered speech in a highly controlled manner. We employ a set of special tokens to specify the locations and types of different stuttering patterns in the source text. When training Stutter-TTS, it is critical to optimize the sampling ratio between fluent and stuttered speech. Stutter-TTS achieves faithful synthesis of utterances with stutter types including phoneme repetition, dysrhythmic phonation, and blocks. In a listening experiment, we find that Stutter-TTS synthesized intended stutter events, as well as non-stuttered words, with high accuracy ($0.63 < F1 < 0.84$). We also demonstrate that fine-tuning an RNN-T ASR model with synthetic stuttered speech improves the recognition of real stuttered speech, with minimal degradation on fluent speech.

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
