# OpenReview forum: "Stutter-TTS:  Controlled Synthesis and Improved Recognition of Stuttered Speech"
_NeurIPS.cc/2022/Workshop/SyntheticData4ML — Neurips 2022 SyntheticData4ML_

### Official Review · Reviewer_KPLk · 2022-10-07
**Simple idea, nicely executed. Lacking in evaluation and application.**

**Rating:** 6
**Confidence:** 4

**Review:**

**Summary and contributions**

   The authors aim to address the problem of inadequate performance of ASR systems on stuttered speech input. For this purpose, they propose a novel TTS system that enables the generation of synthetic speech data by inserting a stutter token into the text input. They perform a subjective qualitative evaluation of the resulting outputs and find that the system generates authentic stuttering patterns.


**Strengths:**


   The proposed method tackles an important problem that limits the accessibility of speech-controlled systems for many people. The authors build their system based on state-of-the-art methods and make sensible decisions for their proposed extensions. The introduction of the stutter token is a simple idea that allows easy control of the location and type of stuttering in a sequence. Results indicate the successful generation of stutter patterns in a synthetic speech sequence.

**Weaknesses:**

   This work is motivated by wanting to improve ASR systems by generating synthetic training data with stuttering.
   However, the paper stops short of this goal.
   A final demonstration of how it improves a standard ASR system would tie this paper together and make this a very nice paper.


**Clarity:**

   The paper follows a clear structure and explains the methodology well.
   Two points that could have been more clear are:

   1. What part of the model/ methodology is novel or an extension of the original architecture?
   2. How the model is trained (on reference stuttering speech) could be hinted at earlier.

**Relation to prior work:**

   The related work is briefly mentioned in the introduction. An additional explanation of the relationship to Sleymanpour et al. would be helpful to understand the novelty of this work.


**Additional Comments.**

   Line 50-51: This sounds confusing. It sounds like Vaswani et al. developed a TTS system (to model stuttering speech). Please clarify.

   Section 3.2, first paragraph: Was this evaluation done on the training data or a held-out data set?

   More details on the training and hyper-parameters would be helpful (in the appendix) to aid reproducibility.

---

### Official Review · Reviewer_7zR1 · 2022-10-17
**Simple approach to address a known problem. Not clear how effective it is.**

**Rating:** 8
**Confidence:** 3

**Review:**

Summary

This paper proposes a simple, yet effective technique where they introduce additional tokens into the source text during training to represent unique stuttering characteristics. By choosing the position of the stutters, the generation model has word-level control of where stuttering occurs in the synthesized utterances.

##########################################################################

Positive points about this paper

Addressing the problem of stutter recognition is an interesting idea. Augmentation is a valid approach to address this lack of data in the ASR task.
The paper covers the literature well and the experiments showcase the usability of the approach. It is well-written and experiments are clearly defined for the most part.

##########################################################################

Concerns and questions:

- Is the evaluation sufficient to support the hypotheses of the paper?

- What is the impact of variations in the generated stutter token to the final output?

---

### Official Review · Reviewer_D2b2 · 2022-10-18
**Interesting paper which provides a solution to generate stuttered speech data**

**Rating:** 7
**Confidence:** 1

**Review:**

The paper presents a transformer based architecture to generate stuttered speech data. The transformer architecture proposed uses in addition a global audio reference encoder along with a noising mechanism at the input layer. The inputs are corrupted by adding additional stutter tokens placed before words. The authors investigate how to mix the synthetic data along with data correpsonding to fluent speech.
I am not an expert in this field, but in my opinion the proposal is reasonable way to generate synthetic speech data. I do not understand Table 2. What is the F1 score being calculated on?

---

### Meta-Review · Area_Chair_JPzM · 2022-10-19

**Recommendation:** Accept